chemical engineering/environmental engineering

lignin model compound, vanillyl alcohol, veratryl alcohol, bleaching, adsorbable organic halogen

**Author for correspondence:**
Shuangquan Yao
e-mail: yaoshuangquan@gxu.edu.cn

# Difference in adsorbable organic halogen formation between phenolic and non-phenolic lignin model compounds in chlorine dioxide bleaching

Lisheng Shi[1,2], Jiayan Ge[1,2], Fuqiang Zhang[1,2], Shuangxi Nie[1,2], Chengrong Qin[1,2] and Shuangquan Yao[1,2]

[1]College of Light Industrial and Food Engineering, Guangxi University, Nanning 530004, People's Republic of China
[2]Guangxi Key Laboratory of Clean Pulp & Papermaking and Pollution Control, Nanning 530004, People's Republic of China

SY, 0000-0003-4982-998X

Adsorbable organic halogen (AOX) is generally formed by the reaction of residual lignin in pulps with chlorine dioxide during bleaching. Lignin has a complex structure. Different functional groups and bonds are present in lignin structures. Phenolic hydroxyl is one of the important functional groups in lignin, and it significantly influences the chemical properties and reactivity. To study the effect of phenolic hydroxyl on AOX formation, vanillyl alcohol (VA) was selected as the phenolic lignin model compound, and veratryl alcohol (VE) was selected as the non-phenolic lignin model compound in this study. The kinetics of AOX formation by the reaction of VA or VE with chlorine dioxide was studied. The effects of pH, chlorine dioxide, lignin model compound concentration and reaction temperature on AOX formation are discussed. The activation energies of the reaction of VA and VE with chlorine dioxide are $16\,242.47\,\text{J}\,\text{mol}^{-1}$ and $281.34\,\text{J}\,\text{mol}^{-1}$, respectively. Thus, we found that the non-phenolic lignin can react with chlorine dioxide to form AOX more easily than phenolic lignin.

## 1. Introduction

During the process of pulp bleaching, the side reactions of HClO and $Cl_2$ in chlorine dioxide solution with residual lignin are

considered to be the source of AOX. AOX cannot be easily degraded; thus, it can remain in the environment for a long time. AOX has adverse effects on plant growth and animal endocrine systems [1].

The structure of residual lignin in unbleached pulp is complex [2]. There are some different extraction processes, such as mechanical, physical, chemical and enzymatic treatments [3,4], to isolate lignin. However, it is difficult to obtain lignin directly by current extraction methods without modifying the lignin structure [5,6]. Therefore, lignin model compounds can be used as a simplified way to study the change in specific lignin structures in related reactions [7]. Lignin model compounds have been widely used in many research fields [8–10]. Scholars have used lignin model compounds to study degradation pathways and reaction mechanisms of different lignin structures [11–13], including the ways in which benzene rings and bonds break during reactions [14–18].

Lignin model compounds have been used in the study of pulp bleaching. Posoknistakul [19] synthesized veratrylglycerol-β-guaiacyl ether (VG), a non-phenolic lignin model compound with a β-O-4 bond. The reactions of VG during hydrogen peroxide bleaching and alkaline oxygen bleaching systems have been studied. The reaction of different lignin structures during oxygen bleaching was studied by Ohmura [20]. It was confirmed that both phenolic and non-phenolic structural units can react with oxygen under simulated oxygen-bleaching conditions. In addition, because some of the non-phenolic components were converted into phenolic components during the oxygen-bleaching reaction, some new phenolic products were detected. Johansson [21] compared the oxidation reactions of different lignin model compounds by oxygen under alkaline oxygen-bleaching conditions. It was found that conjugated side chains of the phenolic lignin model compounds, such as stilbene and enol ethers, can react rapidly, while the reaction rate of propyl-guaiacol and β-aryl ethers is significantly slower. They also showed that the degradation efficiency of lignin model compounds can be accelerated by adding a small amount of hydrogen peroxide during the process of alkaline oxygen bleaching. Using lignin model compounds is one way to study the oxidative degradation of lignin macromolecules in different bleaching systems [22]. The changes in the different lignin structures can be reflected by the reactions of the counterpart lignin model compounds [23,24]. In addition, a delignification kinetic study is an effective method to analyse the degradation mechanism and reaction process of lignin macromolecules [25–27].

Nie [28] studied the oxidation kinetics of a non-phenolic lignin model compound upon reaction with chlorine dioxide using UV–Vis spectroscopy. First, 1-(3,4-dimethoxyphenyl) ethanol (MVA) was selected as the non-phenolic lignin model compound. The results suggested that the oxidation rate of MVA by chlorine dioxide was slow under acidic conditions. A kinetic model of AOX formation during the first bleaching stage ($D_0$) of eucalyptus kraft pulp elemental chlorine free bleaching was established by Yao [29]. The effects of the residual lignin content, chlorine dioxide dose and pH on AOX formation were studied. The degradation and oxidation kinetics of the lignin model compounds, under bleaching conditions, were studied to clarify the reaction mechanism for different lignin structures.

To study the differences in AOX formation [30,31] between phenolic and non-phenolic lignin structures upon reaction with chlorine dioxide during bleaching, the reaction kinetics of phenolic and non-phenolic lignin model compounds, under simulated chlorine dioxide bleaching conditions, were investigated in this study. Vanillyl alcohol (VA) and veratryl alcohol (VE) were selected as the phenolic and non-phenolic lignin model compounds, respectively. The effects of reaction, pH, chlorine dioxide dose and lignin model compound concentration on the AOX formation rate were investigated and compared. The activation energy and the reaction orders were compared. The effect of different lignin structures on AOX formation was analysed. The results provide theoretical guidance for further reduction of AOX and achieve clean bleaching.

# 2. Material and methods

## 2.1. Materials

The chlorine dioxide solution was obtained from a local paper mill (Nanning, China), and the effective chlorine content was 18.51 g l$^{-1}$. VA was selected as the phenolic lignin model compound, and VE was selected as the non-phenolic lignin model compound. VA and VE were purchased in Sigma-Aldrich (Shanghai, China). Other major chemicals and reagents were purchased from Aladdin Biotechnology (Shanghai, China).

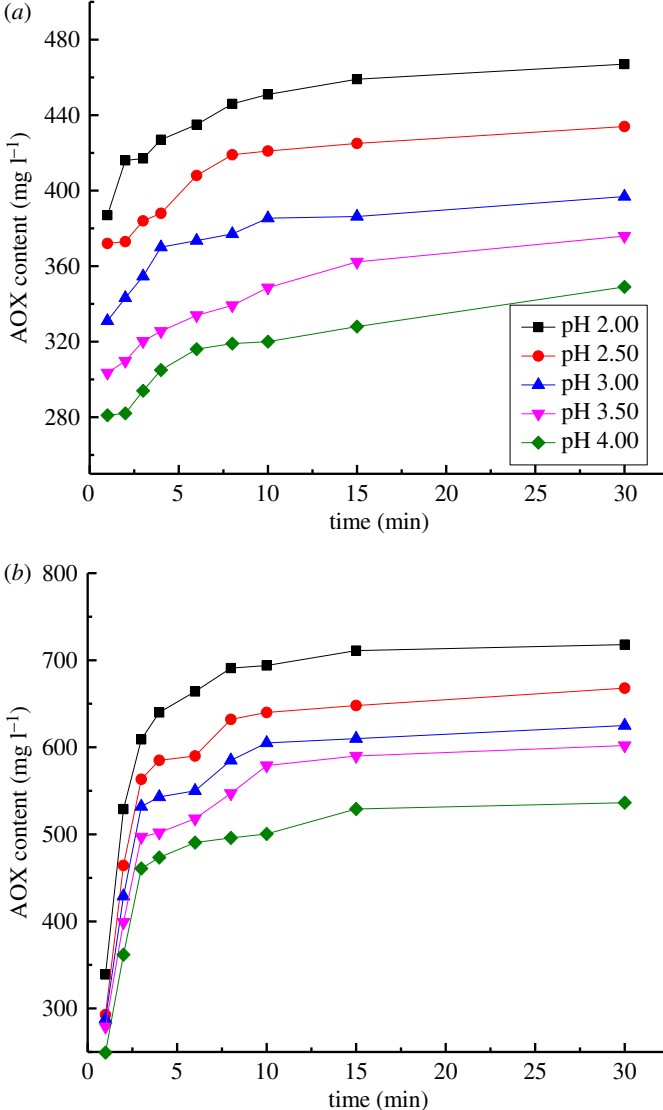

**Figure 1.** Effect of pH on AOX formation. (*a*) Reaction of VA with chlorine dioxide. (*b*) Reaction of VE with chlorine dioxide.

## 2.2. Reaction of chlorine dioxide with lignin model compounds

Chlorine dioxide solution was transferred into a conical bottle, and the pH of the solution was adjusted to 3 with 2 mol l$^{-1}$ H$_2$SO$_4$ at room temperature. Then, a corresponding quantity of VA or VE was added. The reaction was carried out under magnetic stirring (400 rpm) at 343 K. The selection of pH and temperature was based on actual chlorine dioxide bleaching of pulp. The reaction solution was sampled at 1, 2, 3, 4, 6, 8, 10, 15 and 30 min. The samples were stored in a refrigerator, and the AOX content was determined as soon as possible.

## 2.3. AOX analysis

The AOX content of the reaction solution was measured using a multi-X2500 AOX analyser (Jena, Germany). The AOX content was quantified by the micro-coulomb titration method. The basic method and process were described in the previous studies [32–34].

# 3. Results and discussion

## 3.1. Establishment of the kinetic model

The chlorine dioxide dose, lignin model compound concentration, pH and reaction temperature have considerable influence on AOX formation during pulp bleaching. To assess the effect of different

**Table 1.** Linear relationship between log(d$W$/d$t$) and log(H$^+$).

| | VA reacting with chlorine dioxide | | | VE reacting with chlorine dioxide | |
|---|---|---|---|---|---|
| log(H$^+$) | log(d$W$/d$t$) | regression equation of log(d$W$/d$t$) — log(H$^+$) | | log(d$W$/d$t$) | regression equation of log(d$W$/d$t$) — log(H$^+$) |
| −4.00 | 0.70 | $y = 0.07x + 0.98$ | | 1.40 | $y = 0.06x + 1.66$ |
| −3.50 | 0.76 | $R^2 = 0.96$ | | 1.47 | $R^2 = 0.92$ |
| −3.00 | 0.77 | | | 1.48 | |
| −2.50 | 0.82 | | | 1.51 | |
| −2.00 | 0.84 | | | 1.53 | |
| −4.00 | 0.67 | $y = 0.07x + 0.96$ | | 1.38 | $y = 0.06x + 1.64$ |
| −3.50 | 0.74 | $R^2 = 0.96$ | | 1.44 | $R^2 = 0.92$ |
| −3.00 | 0.75 | | | 1.45 | |
| −2.50 | 0.80 | | | 1.49 | |
| −2.00 | 0.82 | | | 1.51 | |
| −4.00 | 0.67 | $y = 0.06x + 0.93$ | | 1.35 | $y = 0.06x + 1.61$ |
| −3.50 | 0.72 | $R^2 = 0.95$ | | 1.42 | $R^2 = 0.91$ |
| −3.00 | 0.73 | | | 1.43 | |
| −2.50 | 0.78 | | | 1.46 | |
| −2.00 | 0.80 | | | 1.48 | |

factors on the AOX formation rate, a kinetic model of AOX formation by using a lignin model substance and reacting it with chlorine dioxide was established by referring to the previous studies [21,28,35]. This model is shown in

$$\frac{\mathrm{d}W}{\mathrm{d}t} = A\mathrm{e}^{-(E/RT)} \cdot [\mathrm{H}^+]^a \cdot [\mathrm{ClO}_2]^b \cdot [\mathrm{C}]^c, \tag{3.1}$$

where $W$ is the content of AOX (mg l$^{-1}$), t is the reaction time (min), d$W$/d$t$ is the AOX formation rate (mg l$^{-1}$ min$^{-1}$), $A$ is the pre-exponential factor, $E$ is the reaction activation energy (J mol$^{-1}$), $R$ is the ideal gas constant (8.314 J mol$^{-1}$ K$^{-1}$), $T$ is the reaction temperature (K), [H$^+$] is the concentration of H$^+$ (mol l$^{-1}$), [ClO$_2$] is the concentration of ClO$_2$ (mol l$^{-1}$), [C] is the concentration of VA or VE (mol l$^{-1}$), $a$, $b$ and $c$ are the reaction orders of H$^+$, ClO$_2$ and the lignin model compound, respectively.

Equation (3.2) was derived from the logarithm of equation (3.1):

$$\log\left(\frac{\mathrm{d}W}{\mathrm{d}t}\right) = A\mathrm{e}^{-(E/RT)} \cdot [\mathrm{H}^+]^a \cdot [\mathrm{ClO}_2]^b \cdot [\mathrm{C}]^c. \tag{3.2}$$

## 3.2. Effect of pH on AOX formation

The effects of pH on AOX formation upon the reaction of VA/VE with chlorine dioxide are shown in figure 1. The reaction conditions were as follows: a reaction temperature of 343 K, a concentration of VA or VE of 62 mmol l$^{-1}$ and a concentration of ClO$_2$ of 124.00 mmol l$^{-1}$. The amount was enough for sufficient reaction progress for VA or VE, which was similar to that during an actual chlorine dioxide bleaching stage. The pH was 2.00, 2.50, 3.00, 3.50 or 4.00.

For the reaction of VA with chlorine dioxide, the AOX formation increased as the pH increased (figure 1$a$). AOX primarily formed within 10 min, which indicated that VA can easily react with chlorine dioxide under acidic conditions. The curve of AOX formation gradually flattened as the reaction proceeded owing to the consumption and decomposition of chlorine dioxide and the oxidation and degradation of VA.

For the reaction of VE with chlorine dioxide, AOX formation was higher at lower pH values (figure 1$b$). The AOX formation rate obviously varied within 10 min, which suggested that AOX can be easily formed by reacting VE with chlorine dioxide at low pH. Owing to the reactions that occurred under acidic conditions, ClO$_2$ converted to HClO in the chlorine dioxide solution at lower pH. In addition, HClO further converted to

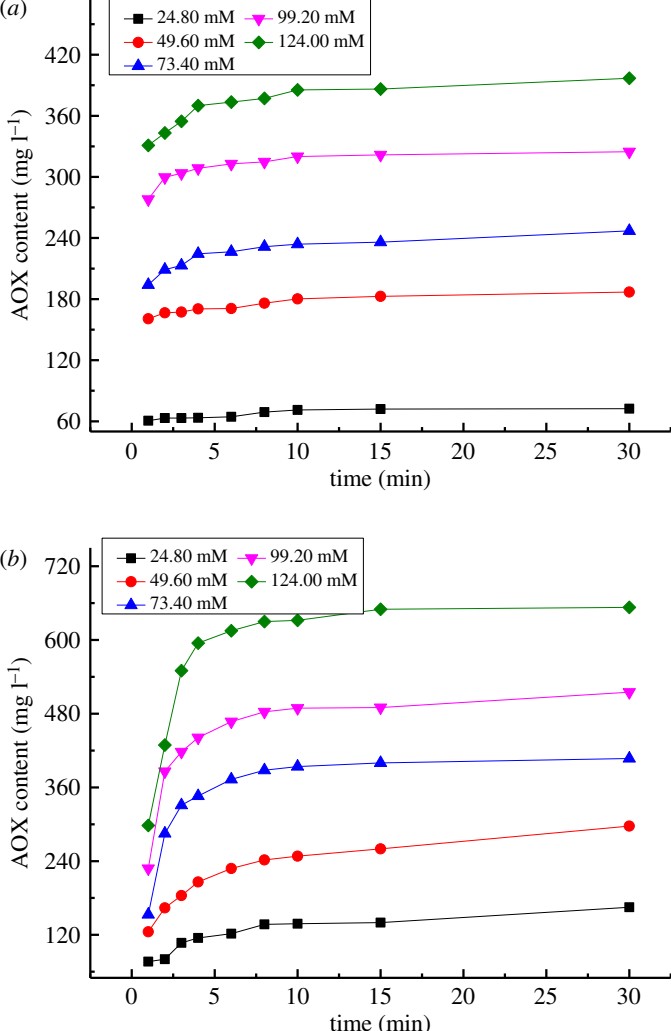

**Figure 2.** Effect of the concentration of ClO$_2$ on AOX formation. (*a*) Reaction of VA with chlorine dioxide. (*b*) Reaction of VE with chlorine dioxide.

Cl$_2$ as the pH decreased below 2.00 [36]. AOX was formed by the side reactions of HClO and Cl$_2$ with residual lignin during chlorine dioxide bleaching. Therefore, there was a considerable amount of HClO and Cl$_2$ that were converted from ClO$_2$ in the chlorine dioxide solution at the beginning of the reactions under low pH conditions. A considerable amount of AOX was formed by the reaction of VE with HClO and Cl$_2$. Meanwhile, HClO and Cl$_2$ were rapidly consumed and ClO$_2$ was gradually decomposed. The curve for AOX formation was thus stable.

Other conditions were maintained, when only the pH was changed. Equation (3.3) can be obtained by rearranging equation (3.2):

$$\log\left(\frac{\mathrm{d}W}{\mathrm{d}t}\right) = \log(A\mathrm{e}^{-(E/RT)} \cdot [\mathrm{ClO_2}]^b \cdot [\mathrm{C}]^c) + a\log(\mathrm{H^+}). \tag{3.3}$$

According to the data from the experiments, the linear relationship between $\log(\mathrm{d}W/\mathrm{d}t)$ and $\log(\mathrm{H^+})$ was determined by the linear regression analysis. The regression equations for $\log(\mathrm{d}W/\mathrm{d}t) - \log(\mathrm{H^+})$ are shown in table 1 for the reactions of VA or VE with chlorine dioxide. The value of parameter $a$ was thus determined. For the reaction of VA with chlorine dioxide, the slopes of the three fitting equations were 0.07, 0.07 and 0.06. Parameter a1 in the kinetic model of the reaction of VA with chlorine dioxide to form AOX was determined by averaging these three values; thus, $a_1$ was determined to be 0.07. For the reaction of VE with chlorine dioxide, the slopes of the three fitting equations were 0.06, 0.06, and 0.06. Parameter $a_2$ in the kinetic model of the reaction of VE with chlorine dioxide to form AOX was determined by averaging these three values; thus, $a_2$ was determined to be 0.06. The reaction orders of the proton are 0.07 and 0.06. The values show that the acidities of the reaction solutions did not affect

**Table 2.** Linear relationship between log(d$W$/d$t$) and log(ClO$_2$).

| | VA reacting with chlorine dioxide | | | VE reacting with chlorine dioxide | |
|---|---|---|---|---|---|
| log(ClO$_2$) | log(d$W$/d$t$) | regression equation of log(d$W$/d$t$) − log(ClO$_2$) | | log(d$W$/d$t$) | regression equation of log(d$W$/d$t$) − log(ClO$_2$) |
| −1.61 | 0.10 | $y = 0.94x + 1.61$ | | 0.79 | $y = 0.87x + 2.20$ |
| −1.30 | 0.34 | $R^2 = 0.96$ | | 1.07 | $R^2 = 0.97$ |
| −1.13 | 0.62 | | | 1.28 | |
| −1.00 | 0.60 | | | 1.29 | |
| −0.91 | 0.77 | | | 1.40 | |
| −1.61 | 0.06 | $y = 0.94x + 1.56$ | | 0.75 | $y = 0.85x + 2.13$ |
| −1.30 | 0.30 | $R^2 = 0.96$ | | 1.02 | $R^2 = 0.98$ |
| −1.13 | 0.57 | | | 1.22 | |
| −1.00 | 0.56 | | | 1.24 | |
| −0.91 | 0.73 | | | 1.35 | |
| −1.61 | −0.01 | $y = 0.93x + 1.48$ | | 0.70 | $y = 0.82x + 2.04$ |
| −1.30 | 0.23 | $R^2 = 0.96$ | | 0.97 | $R^2 = 0.98$ |
| −1.13 | 0.50 | | | 1.15 | |
| −1.00 | 0.48 | | | 1.17 | |
| −0.91 | 0.65 | | | 1.28 | |

AOX formation. However, the activity of chlorine dioxide was affected by acidity ·ClO in chlorine dioxide solution. It was converted to HClO and Cl$_2$ when the value of pH was low. As a consequence, the AOX formation increased as pH value decreased. This corresponds to the results shown in figure 1.

## 3.3. Effect of chlorine dioxide dose on AOX formation

The effects of chlorine dioxide dose on AOX formation upon the reactions of VA/VE with chlorine dioxide are shown in figure 2. The reaction conditions are as follows: reaction temperature of 343 K, a pH of 3.00, a concentration of VA or VE of 62.00 mmol l$^{-1}$ and a concentration of ClO$_2$ of 24.80, 49.60, 73.40, 99.20 or 124.00 mmol l$^{-1}$.

The AOX formed by reacting VA with chlorine dioxide significantly increased owing to the increase in the concentration of ClO$_2$, as shown in figure 2$a$. As the concentration of ClO$_2$ increased from 24.80 to 49.60 mmol l$^{-1}$, the AOX formation increased by approximately three times. This indicated that the dosage of chlorine dioxide is an important factor that affects AOX formation. The curve of AOX formation is relatively smooth under low ClO$_2$ concentration conditions owing to the quantities of HClO and Cl$_2$ being much smaller in the chlorine dioxide solution. Within 5 min of the reaction of VA with chlorine dioxide, HClO and Cl$_2$ reacted with VA rapidly to form AOX, and they were consumed completely. The rate of AOX formation changed considerably with the increase in chlorine dioxide dosage after 5 min of reaction time.

For the reaction of VE with chlorine dioxide, the AOX formation significantly increased as the concentration of chlorine dioxide increased (figure 2$b$). The AOX formation rate of the reaction of VE with chlorine dioxide was faster than that of VA with chlorine dioxide. With an increase in the concentration of ClO$_2$, the AOX formation rate was promoted. AOX was primarily formed within 5 min.

Other conditions were maintained, and only the concentration of ClO$_2$ was changed. Equation (3.4) can be obtained by rearranging equation (3.2):

$$\log\left(\frac{\mathrm{d}W}{\mathrm{d}t}\right) = \log(A\mathrm{e}^{-(E/RT)}[\mathrm{H}^+]^a \cdot [\mathrm{C}]^c) + b\log(\mathrm{ClO}_2). \tag{3.4}$$

The regression equations for log(d$W$/d$t$) − log(ClO$_2$) were determined by linear fitting of the data (table 2). Parameter $b$ was determined by the slope of the regression equations of log(d$W$/d$t$) − log

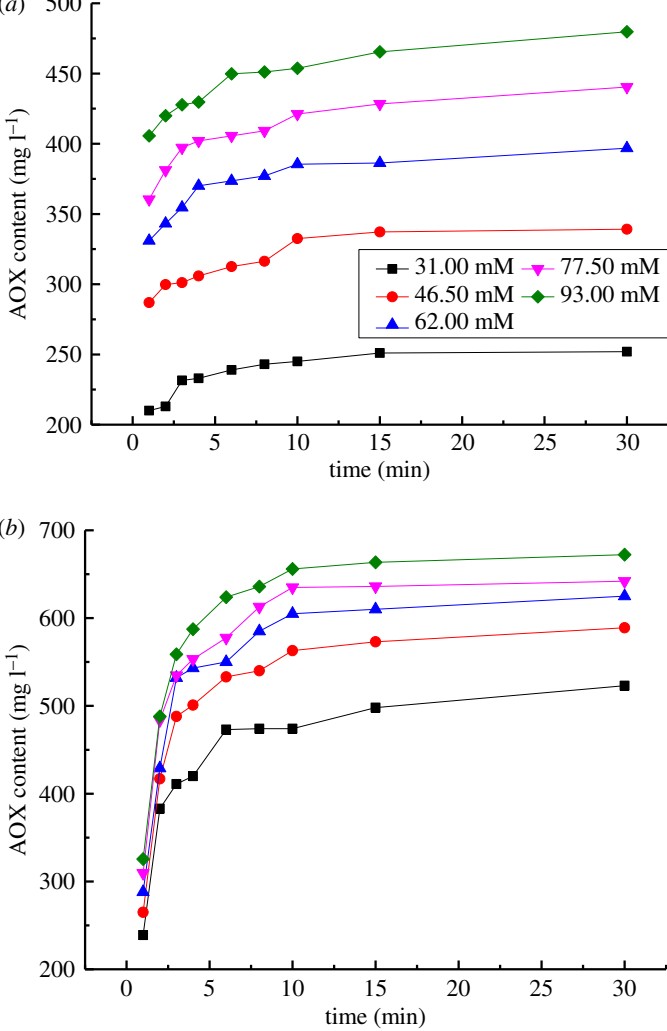

**Figure 3.** Effect of concentration of lignin model compound on AOX formation. (*a*) Reaction of VA with chlorine dioxide. (*b*) Reaction of VE with chlorine dioxide.

($ClO_2$). For the reaction of VA with chlorine dioxide, the slopes of the three fitting equations were 0.94, 0.94 and 0.93. Parameter $b_1$ in the kinetic model of the reaction of VA with chlorine dioxide to form AOX was determined by averaging these three values, and it was $b_1 = 0.94$. For the reaction of VE with chlorine dioxide, the slopes of the three fitting equations were 0.87, 0.85 and 0.82. Parameter $b_2$ in the kinetic model for the reaction of VE with chlorine dioxide to form AOX was determined by averaging these three values, and it was $b_2 = 0.85$.

## 3.4. Effect of lignin model compound concentration on AOX formation

The effects of the concentration of VA or VE on AOX formation by the reaction of VA/VE with chlorine dioxide are shown in figure 3. The reaction conditions were as follows: reaction temperature of 343 K, a pH of 3.00, a concentration of $ClO_2$ of 124.00 mmol l$^{-1}$ and a concentration of VA or VE of 31.00, 46.50, 62.00, 77.50 or 93.00 mmol l$^{-1}$.

As shown in figure 3*a*, for the reaction of VA with chlorine dioxide, the AOX formation rate was slower when the concentration of VA was below 62.00 mmol l$^{-1}$ for the first 10 min, and it increased with increasing VA concentration. This suggested that the concentration of VA could significantly affect the AOX formation. For the reaction of VE with chlorine dioxide, the concentration of VE also had a significant effect on the AOX formation rate. The AOX formation rate was remarkably higher during the first 2 min. Thus, the rate and content of AOX formed by reacting VE with chlorine dioxide were higher than those of VA at the same concentration for the lignin model compound.

**Table 3.** Linear relationship between log(dW/dt) and log(C).

| | VA reacting with chlorine dioxide | | VE reacting with chlorine dioxide | |
|---|---|---|---|---|
| $\log(C)$ | $\log(\mathrm{d}W/\mathrm{d}t)$ | regression equation of $\log(\mathrm{d}W/\mathrm{d}t) - \log(C)$ | $\log(\mathrm{d}W/\mathrm{d}t)$ | regression equation of $\log(\mathrm{d}W/\mathrm{d}t) - \log(C)$ |
| −1.51 | 0.52 | $y = 0.33x + 1.03$ | 1.35 | $y = 0.30x + 1.82$ |
| −1.33 | 0.59 | $R^2 = 0.91$ | 1.45 | $R^2 = 0.91$ |
| −1.21 | 0.65 | | 1.48 | |
| −1.11 | 0.68 | | 1.48 | |
| −1.03 | 0.66 | | 1.50 | |
| −1.51 | 0.46 | $y = 0.35x + 1.01$ | 1.33 | $y = 0.30x + 1.80$ |
| −1.33 | 0.54 | $R^2 = 0.93$ | 1.42 | $R^2 = 0.91$ |
| −1.21 | 0.60 | | 1.45 | |
| −1.11 | 0.63 | | 1.46 | |
| −1.03 | 0.61 | | 1.48 | |
| −1.51 | 0.39 | $y = 0.39x + 0.98$ | 1.31 | $y = 0.29x + 1.77$ |
| −1.33 | 0.47 | $R^2 = 0.94$ | 1.40 | $R^2 = 0.91$ |
| −1.21 | 0.53 | | 1.43 | |
| −1.11 | 0.57 | | 1.43 | |
| −1.03 | 0.56 | | 1.46 | |

Other conditions were maintained, and only the concentration of VA or VE was changed. Equation (3.5) could be obtained by rearranging equation (3.2):

$$\log\left(\frac{\mathrm{d}W}{\mathrm{d}t}\right) = \log(A\mathrm{e}^{-(E/RT)} \cdot [\mathrm{H}^+]^a \cdot [\mathrm{ClO}_2]^b) + c\log(C). \tag{3.5}$$

The linear fitting equations of $\log(\mathrm{d}W/\mathrm{d}t) - \log(C)$ were determined by linear regression analysis of the experimental results (table 3). Parameter $c$ was determined by the slopes of the $\log(\mathrm{d}W/\mathrm{d}t) - \log(C)$ regression equations. For the reaction of VA with chlorine dioxide, the slopes of the three fitting equations were 0.33, 0.35 and 0.39. Parameter $c_1$ in the kinetic model of the reaction of VA with chlorine dioxide to form AOX was determined by averaging these three values, and $c_1 = 0.36$. For the reaction of VE with chlorine dioxide, the slopes of the three fitting equations were 0.30, 0.30 and 0.29. Parameter $c_2$ in the kinetic model of the reaction of VE with chlorine dioxide to form AOX was determined by averaging these three values, and $c_2 = 0.30$.

## 3.5. Effect of reaction temperature on AOX formation

The effects of reaction temperature on AOX formation by the reaction of VA/VE with chlorine dioxide are shown in figure 4. The reaction conditions were as follows: a pH of 3.00, a concentration of ClO2 of 124.00 mmol l$^{-1}$, a concentration of VA or VE of 62.00 mmol l$^{-1}$ and a reaction temperature of 323, 328, 333, 338 or 343 K.

According to the Arrhenius equation, the reaction rate constant, $k$, can be obtained:

$$k = A\mathrm{e}^{-(E/RT)}. \tag{3.6}$$

Equation (3.7) was derived from the logarithm of equation (3.6):

$$\log(k) = \log(A) - \frac{E}{RT} \cdot \log e. \tag{3.7}$$

Equation (3.8) was derived from equations (3.2) and (3.7):

$$\log(k) = \log\left(\frac{\mathrm{d}W}{\mathrm{d}t}\right) - a \cdot [\mathrm{H}^+] - b \cdot [\mathrm{ClO}_2] - c \cdot [\mathrm{C}]. \tag{3.8}$$

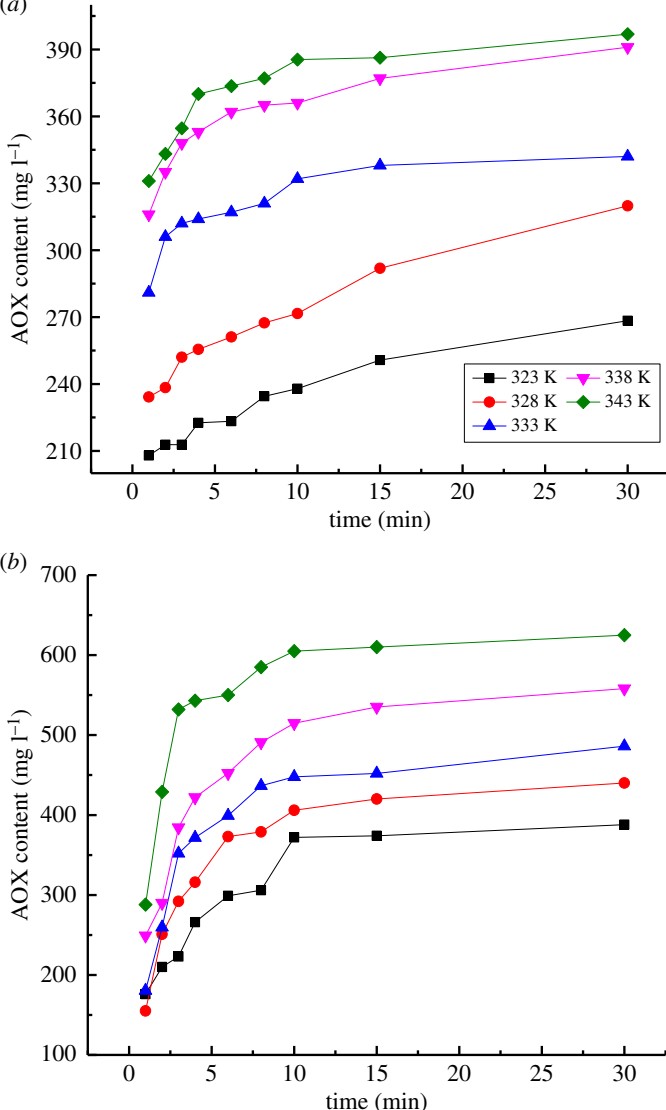

**Figure 4.** Effect of temperature on AOX formation. (*a*) Reaction of VA with chlorine dioxide. (*b*) Reaction of VE with chlorine dioxide.

Parameters $a_1$, $b_1$ and $c_1$ and the concentrations of $H^+$, $ClO_2$ and VA were included in equation (3.8) to calculate log (*k*). Linear fitting equations for $\log(k) - 1/T$ were obtained by linear regression analysis of the experimental results (table 4). For the reaction of VA with chlorine dioxide, the average of the intercepts of the linear regression equations was determined to be 4.87, log ($A_1$) = 4.87. The pre-exponential factor was obtained; $A_1$ = 73 824.40. The slopes of the linear regression equations were −899.55, −849.89 and −795.92. The average of the slopes was −848.45. The reaction activation energy of VA with chlorine dioxide was thus determined to be $E_1$ = 16 242.47 J mol$^{-1}$, which indicated that the activation energy of the reaction of VA with chlorine dioxide was small, and AOX was quickly formed.

The kinetic model of AOX formation from the reaction of VA with chlorine dioxide is shown as

$$\frac{dW}{dt} = 73824.40 \cdot e^{-(-16242.47/RT)} \cdot [H^+]^{0.07} \cdot [ClO_2]^{0.94} \cdot [C]^{0.36}. \tag{3.9}$$

In the kinetic model, the reaction orders of $H^+$, $ClO_2$ and VA are not integers, which indicates that the AOX formation from the reaction of VA with chlorine dioxide is a complex multivariate reaction. The reaction order of $H^+$ is significantly small; thus, $H^+$ may play a catalytic role in the reaction.

Parameters $a_2$, $b_2$ and $c_2$ and the concentrations of $H^+$, $ClO_2$ and VE were included in equation (3.8) to calculate the log (*k*). For the reaction of VE with chlorine dioxide, the average intercept of the linear regression equations was 3.01, log ($A_2$) = 3.01. The pre-exponential factor was determined to be $A_2$ = 1012.05. The slopes of the linear regression equations were −14.70, −14.54 and −14.37, respectively.

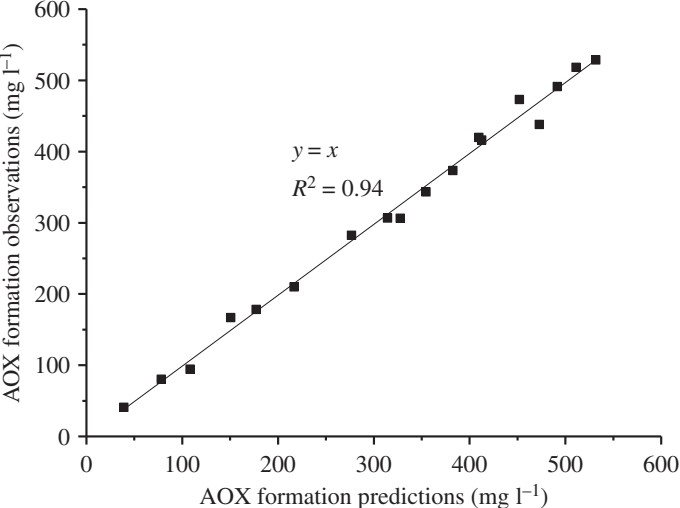

**Figure 5.** Linear relationship between the predictions and experimental data for the reaction of VA with chlorine dioxide.

**Table 4.** Linear relationship between log($k$) and 1/$T$ in the reaction of VE with chlorine dioxide.

| 1/$T$ ($\times10^{-3}$) | VA reacting with chlorine dioxide | | VE reacting with chlorine dioxide | |
| | log($k$) | regression equation of log($k$) — 1/$T$ | log($k$) | regression equation of log($k$) — 1/$T$ |
| --- | --- | --- | --- | --- |
| 2.92 | 2.24 | $y = -899.55x + 4.87$ | 2.79 | $y = -14.70x + 4.67$ |
| 2.99 | 2.21 | $R^2 = 0.98$ | 2.78 | $R^2 = 0.97$ |
| 3.00 | 2.16 | | 2.76 | |
| 3.05 | 2.14 | | 2.74 | |
| 3.10 | 2.07 | | 2.71 | |
| 2.92 | 2.22 | $y = -849.89x + 4.70$ | 2.77 | $y = -14.54x + 4.65$ |
| 2.96 | 2.19 | $R^2 = 0.97$ | 2.76 | $R^2 = 0.96$ |
| 3.00 | 2.14 | | 2.74 | |
| 3.05 | 2.13 | | 2.72 | |
| 3.10 | 2.06 | | 2.69 | |
| 2.92 | 2.19 | $y = -795.92x + 4.52$ | 2.74 | $y = -14.37x + 4.62$ |
| 2.96 | 2.17 | $R^2 = 0.94$ | 2.74 | $R^2 = 0.95$ |
| 3.00 | 2.12 | | 2.71 | |
| 3.05 | 2.11 | | 2.69 | |
| 3.10 | 2.04 | | 2.66 | |

The average of the slopes was −14.54. The reaction activation energy of VE with chlorine dioxide was determined to be $E_2 = 281.34$ J mol$^{-1}$, which is significantly lower than the reaction activation energy of VA with chlorine dioxide, which indicated that VE can easily react with chlorine dioxide to form AOX. This was because the hydrogen atoms on the benzene ring of VE were prone to be replaced by chlorine atoms to form AOX. However, oxidation and degradation of benzene ring were the dominant reactions when chlorine dioxide reacted with VA.

The kinetic model of AOX formation by the reaction of VE with chlorine dioxide is shown in

$$\frac{\mathrm{d}W}{\mathrm{d}t} = 1012.05 \cdot \mathrm{e}^{-(281.34/RT)} \cdot [\mathrm{H}^+]^{0.06} \cdot [\mathrm{ClO}_2]^{0.98} \cdot [\mathrm{C}]^{0.91} . \tag{3.10}$$

For the reaction of VE with chlorine dioxide, the concentration of H$^+$ also had little effect on AOX formation, whereas the concentration of ClO$_2$ significantly affected the AOX formation. In addition,

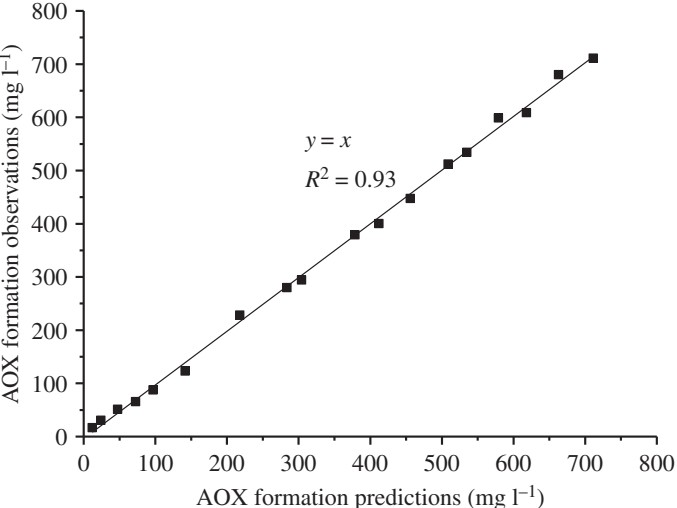

**Figure 6.** Linear relationship between the predictions and experimental data for the reaction of VE with chlorine dioxide.

the effect of the concentration of VE on AOX formation was more remarkable than that of VA, which suggested that in an actual bleaching process, increasing the proportion of non-phenolic structures in the lignin may promote AOX formation during chlorine dioxide bleaching.

### 3.6. Kinetic model accuracy analysis

The predicted AOX formation content from the reaction of lignin model compounds with chlorine dioxide could be calculated by the established kinetic model. For the reaction of VA with chlorine dioxide, the relationship between the predicted content of AOX formation and the experimental data is shown in figure 5. The $R^2$ of the fitting equation was 0.94, which indicated that the kinetic model was in a good agreement with the experimental results. For the reaction of VE with chlorine dioxide, the relationship between the predicted content of AOX formation and the experimental data is shown in figure 6. The $R^2$ of the fitting equation was 0.93, and the kinetic model was able to effectively predict AOX formation.

## 4. Conclusion

The effects of $ClO_2$, pH, temperature and lignin model compound concentration on the AOX formation rate were studied. A kinetic model for AOX formation upon the reaction of VA/VE with chlorine dioxide was established. The activation energies for the reaction of VA or VE with chlorine dioxide were obtained. The results showed that the reaction activation energy for VE with chlorine dioxide was significantly lower than that of VA with chlorine dioxide, which indicated that the non-phenolic lignin model compound can react with chlorine dioxide to form AOX more easily than the phenolic lignin model compound.

Data accessibility. The datasets supporting this article have been uploaded as part of the manuscript and electronic supplementary material.
Authors' contributions. S.Y. designed the study. L.S., J.G., F.Z., S.N. and C.Q. collected all data for analysis. L.S. and S.Y. analysed the data, interpreted the results and wrote the manuscript. All authors gave final approval for publication.
Competing interests. We declare we have no competing interests.
Funding. This project was sponsored by the National Natural Science Foundation of China (grant no. 31760192).
Acknowledgements. We thank Guangxi Key Laboratory of Clean Pulp & Papermaking and Pollution Control for their help.

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
