## [Reviewer comments · Royal Society Open Science]

Review History

RSOS-191202.R0 (Original submission)

Review form: Reviewer 1

Is the manuscript scientifically sound in its present form?

Yes

Are the interpretations and conclusions justified by the results?

Yes

Is the language acceptable?

Yes

Do you have any ethical concerns with this paper?

No

Have you any concerns about statistical analyses in this paper?

No

Recommendation?

Accept with minor revision (please list in comments)

Comments to the Author(s)

In this work the authors present their results on the effects of lignin structure on adsorbable organic halogen (AOX) formation. Two important components of lignin, phenolic lignin and non-phenolic lignin, were studied. The results show that AOX was more easily generated from the reaction of non-phenolic lignin and chlorine dioxide. I believe the manuscript can be accepted after minor revision. Some specific points are showed below.

1. Introduction. The readers are confused about the structure and meaning of the article. This description is missing from the introduction, it needs to be added.
2. Introduction. "It is difficult to obtain lignin directly by current extraction methods without modifying the lignin structure". Readers want to know the methods and characteristics of lignin extraction.
3. "The pH was 2.00, 2.50, 3.00, 3.50, or 4.00". Readers want to know how these pH values are selected. Whether it is close to the actual bleaching experiment?
4. Equations (9) and (10): The reaction orders of proton are 0.07 and 0.06, respectively. These values show that the acidities of the reaction solutions did not affect the AOX formation, which are not in accordance with the results shown in Fig. 1. This is because the stability of chlorine dioxide solution is affected by pH. The authors need to explain clearly in the article.
5. The bond between structural units of lignin has a great influence on lignin properties. The authors may consider to carry on the thorough research from this aspect.

Review form: Reviewer 2

Is the manuscript scientifically sound in its present form?

Yes

Are the interpretations and conclusions justified by the results?

Yes

Is the language acceptable?

Yes

Do you have any ethical concerns with this paper?

No

Have you any concerns about statistical analyses in this paper?

No

Recommendation?

Accept with minor revision (please list in comments)

Comments to the Author(s)

The manuscript RSOS-191202 by Yao and coworkers described the difference of phenolic and nonphenolic lignin-derived compounds in adsorbable organic halogen (AOX). Detailed kinetic data associated to substrate/reactant concentrations and temperature were obtained, from which two regression equations were generalized. I think this manuscript is suitable for publishing on Royal Society Open Science after minor revisions.

1. Do the vanillyl alcohol (VA) and veratryl alcohol (VE) follow the same reaction pathways on the treatment of chlorine dioxide? The explanations on the difference activation energies between VA and VE should be added in the manuscript.
2. The models, vanillyl alcohol (VA) and veratryl alcohol (VE) could be used for lignin research, but they cannot represent lignin really. Therefore, the description of "non-phenolic lignin" in summary and "non-phenolic lignin structure" in conclusion should be modified.
3. Some literature on the reactivity study of phenolic and nonphenolic lignin models should be cited, such as *Angew. Chem. Int. Ed.* 2012, 51, 3410 and *ACS Catal.* 2019, 9, 4054.

Decision letter (RSOS-191202.R0)

08-Sep-2019

Dear Dr Yao

On behalf of the Editors, I am pleased to inform you that your Manuscript RSOS-191202 entitled "Difference in adsorbable organic halogen formation between phenolic and non-phenolic lignin model compounds in chlorine dioxide bleaching" has been accepted for publication in Royal Society Open Science subject to minor revision in accordance with the referee suggestions. Please find the referees' comments at the end of this email.

The reviewers and handling editors have recommended publication, but also suggest some minor revisions to your manuscript. Therefore, I invite you to respond to the comments and revise your manuscript.

- Ethics statement

- Data accessibility

<http://datadryad.org/submit?journalID=RSOS&manu=RSOS-191202>

- **Competing interests**

- **Authors' contributions**

- **Acknowledgements**

- **Funding statement**

Because the schedule for publication is very tight, it is a condition of publication that you submit the revised version of your manuscript before 17-Sep-2019. Please note that the revision deadline will expire at 00.00am on this date. If you do not think you will be able to meet this date please let me know immediately.

on behalf of Dr Silvia Vignolini (Associate Editor) and R. Kerry Rowe (Subject Editor)
openscience@royalsociety.org

Reviewer comments to Author:

Reviewer: 1

Comments to the Author(s)

In this work the authors present their results on the effects of lignin structure on adsorbable organic halogen (AOX) formation. Two important components of lignin, phenolic lignin and non-phenolic lignin, were studied. The results show that AOX was more easily generated from the reaction of non-phenolic lignin and chlorine dioxide. I believe the manuscript can be accepted after minor revision. Some specific points are showed below.

1. Introduction. The readers are confused about the structure and meaning of the article. This description is missing from the introduction, it needs to be added.
2. Introduction. "It is difficult to obtain lignin directly by current extraction methods without modifying the lignin structure". Readers want to know the methods and characteristics of lignin extraction.
3. "The pH was 2.00, 2.50, 3.00, 3.50, or 4.00". Readers want to know how these pH values are selected. Whether it is close to the actual bleaching experiment?
4. Equations (9) and (10): The reaction orders of proton are 0.07 and 0.06, respectively. These values show that the acidities of the reaction solutions did not affect the AOX formation, which are not in accordance with the results shown in Fig. 1. This is because the stability of chlorine dioxide solution is affected by pH. The authors need to explain clearly in the article.
5. The bond between structural units of lignin has a great influence on lignin properties. The authors may consider to carry on the thorough research from this aspect.

Reviewer: 2

Comments to the Author(s)

The manuscript RSOS-191202 by Yao and coworkers described the difference of phenolic and nonphenolic lignin-derived compounds in adsorbable organic halogen (AOX). Detailed kinetic data associated to substrate/reactant concentrations and temperature were obtained, from which two regression equations were generalized. I think this manuscript is suitable for publishing on Royal Society Open Science after minor revisions.

1. Do the vanillyl alcohol (VA) and veratryl alcohol (VE) follow the same reaction pathways on the treatment of chlorine dioxide? The explanations on the difference activation energies between VA and VE should be added in the manuscript.
2. The models, vanillyl alcohol (VA) and veratryl alcohol (VE) could be used for lignin research, but they cannot represent lignin really. Therefore, the description of "non-phenolic lignin" in summary and "non-phenolic lignin structure" in conclusion should be modified.
3. Some literature on the reactivity study of phenolic and nonphenolic lignin models should be cited, such as *Angew. Chem. Int. Ed.* 2012, 51, 3410 and *ACS Catal.* 2019, 9, 4054.

Author's Response to Decision Letter for (RSOS-191202.R0)

See Appendix A.

Decision letter (RSOS-191202.R1)

20-Sep-2019

Dear Dr Yao,

I am pleased to inform you that your manuscript entitled "Difference in adsorbable organic halogen formation between phenolic and non-phenolic lignin model compounds in chlorine dioxide bleaching" is now accepted for publication in Royal Society Open Science.

Please note that the email address qinchengrong@sina.com is not receiving our messages - please ensure that you supply the editorial office with an accurate email address for your colleague.

on behalf of Dr Silvia Vignolini (Associate Editor) and R. Kerry Rowe (Subject Editor)
openscience@royalsociety.org

Appendix A

Dear Reviewers,

Thank you for your letter and for the comments concerning our manuscript entitled “Difference in adsorbable organic halogen formation between phenolic and non-phenolic lignin model compounds in chlorine dioxide bleaching” (Manuscript Number: RSOS-191202). We have studied your comments carefully and have made corrections which we hope could meet your requirements. All changes have been highlighted in the revised version (red highlighting).

Questions you put forward are explained as follows:

Reviewer: 1

1. Introduction. The readers are confused about the structure and meaning of the article. This description is missing from the introduction, it needs to be added.

To study the differences in AOX formation between phenolic and non-phenolic lignin structures upon reaction with chlorine dioxide during bleaching, the reaction kinetics of phenolic and non-phenolic lignin model compounds under simulated chlorine dioxide bleaching conditions were investigated in this study. Vanillyl alcohol (VA) and veratryl alcohol (VE) were selected as the phenolic and non-phenolic lignin model compounds, respectively. The effects of reaction, pH, chlorine dioxide dose, and lignin model compound concentration on the AOX formation rate were investigated and compared. The activation energy and the reaction orders were compared. The effect of different lignin structures on AOX formation was analyzed. The results provide theoretical guidance for further reduction of AOX and achieve clean bleaching.

2. Introduction. “It is difficult to obtain lignin directly by current extraction methods without modifying the lignin structure”. Readers want to know the methods and characteristics of lignin extraction.

There are some different extraction processes to isolate lignin, such as mechanical, physical, chemical and enzymatic treatment.

3. “The pH was 2.00, 2.50, 3.00, 3.50, or 4.00”. Readers want to know how these pH values are selected. Whether it is close to the actual bleaching experiment?

The selection of pH was based on the pH conditions during chlorine dioxide bleaching of pulp. In addition, reacting conditions of the experiments, such as temperature, were carrying on under simulated chlorine dioxide bleaching of pulp. It has been explained further in the manuscript.

4. Equations (9) and (10): The reaction orders of proton are 0.07 and 0.06, respectively. These values show that the acidities of the reaction solutions did not affect the AOX formation, which are not in accordance with the results shown in Fig. 1. This is because the stability of chlorine dioxide

solution is affected by pH. The authors need to explain clearly in the article.

The activity of chlorine dioxide was affected by acidity $\cdot\text{ClO}$ in chlorine dioxide solution. It was converted to HClO and Cl_2 when the value of pH was low. As consequence, the AOX formation increased as pH value decreasing. The manuscript has been modified as requested. The explanations have been added in the revised version.

5. The bond between structural units of lignin has a great influence on lignin properties. The authors may consider to carry on the thorough research from this aspect.

Thanks for your advice. Some studies about the influence of the bonds between structural units of lignin on AOX formation are carrying on by our team.

Reviewer: 2

1. Do the vanillyl alcohol (VA) and veratryl alcohol (VE) follow the same reaction pathways on the treatment of chlorine dioxide? The explanations on the difference activation energies between VA and VE should be added in the manuscript.

Different lignin models have different reaction pathways on the treatment of chlorine dioxide. The effect of different lignin structures on AOX formation was analyzed. The explanations have been added in the manuscript as requested.

2. The models, vanillyl alcohol (VA) and veratryl alcohol (VE) could be used for lignin research, but they cannot represent lignin really. Therefore, the description of “non-phenolic lignin” in summary and “non-phenolic lignin structure” in conclusion should be modified.

The manuscript has been modified as requested.

3. Some literature on the reactivity study of phenolic and nonphenolic lignin models should be cited, such as *Angew. Chem. Int. Ed.* 2012, 51, 3410 and *ACS Catal.* 2019, 9, 4054.

The related literature has been added in the revised version.

As a whole, issues the reviewers suggested are very pertinent, which are very helpful to modified my entire paper and thank you very much again.

Yours Sincerely
The author
Shuangquan Yao